# Genetic Analysis of an F_2_ Population Derived from the Cotton Landrace Hopi Identified Novel Loci for Boll Glanding

**DOI:** 10.3390/ijms25137080

**Published:** 2024-06-27

**Authors:** Avinash Shrestha, Junghyun Shim, Puneet Kaur Mangat, Lakhvir Kaur Dhaliwal, Megan Sweeney, Rosalyn B. Angeles-Shim

**Affiliations:** 1Department of Plant and Soil Science, Davis College of Agricultural Sciences and Natural Resources, Texas Tech University, Lubbock, TX 79409, USA; avishres@ttu.edu (A.S.); junghyun.shim@ttu.edu (J.S.); puneet.mangat@ttu.edu (P.K.M.); lakhvir.kaur@ttu.edu (L.K.D.); 2BASF Corporation, 407 Davis Drive, Morrisville, NC 27560, USA; megan.sweeney@basf.com

**Keywords:** cottonseed, glanding, gossypol, genetic diversity, transcription factors, QTL mapping

## Abstract

Landraces are an important reservoir of genetic variation that can expand the narrow genetic base of cultivated cotton. In this study, quantitative trait loci (QTL) analysis was conducted using an F_2_ population developed from crosses between the landrace Hopi and inbred TM-1. A high-density genetic map spanning 2253.11 and 1932.21 cM for the A and D sub-genomes, respectively, with an average marker interval of 1.14 cM, was generated using the CottonSNP63K array. The linkage map showed a strong co-linearity with the physical map of cotton. A total of 21 QTLs were identified, controlling plant height (1), bract type (1), boll number (1), stem color (2), boll pitting (2), fuzz fiber development (2), boll shape (3), boll point (4), and boll glanding (5). In silico analysis of the novel QTLs for boll glanding identified a total of 13 candidate genes. Analysis of tissue-specific expression of the candidate genes suggests roles for the transcription factors *bHLH1*, *MYB2*, and *ZF1* in gland formation. Comparative sequencing of open reading frames identified early stop codons in all three transcription factors in Hopi. Functional validation of these genes offers avenues to reduce glanding and, consequently, lower gossypol levels in cottonseeds without compromising the defense mechanisms of the plant against biotic stresses.

## 1. Introduction

Cultivated cotton is one of the leading agricultural commodities in the global economy. The crop is primarily cultivated for its natural fiber, but higher turnover rates have been reported for cottonseed byproducts such as oil (16–25%) and meal (50%) compared to lint (7%) [1].

Cotton oil, which is considered “Heart Oil” by the American Heart Association, is approximately 50% polyunsaturated and is rich in vitamin E. Its better nutritional value of 9 kcal/g and higher digestibility of 97% compared to soybean, safflower, and sunflower oil make it a healthier edible oil option [2]. Cottonseed meal, on the other hand, is a rich source of proteins for older cattle and fish [2].

Despite the nutritional and economic benefits of cotton oil and protein meals, the widespread production and utilization of these byproducts have been limited by the presence of gossypol in cottonseeds. Cotton plants have lysigenous pigment glands that contain a diverse group of terpenoids such as gossypol, hemigossypolone, and helicoides H_1_, H_2_, H_3_, and H_4_. Hemigossypolone and helicoides are the major terpenoids found in the leaves, whereas gossypol is the major terpenoid produced in flowers and seeds [3]. Gossypol is toxic to major arthropod pests and therefore acts as a natural defense mechanism against cotton bollworms, aphids, and *Heliothis virescens* [4]. This toxicity extends to monogastric animals and has been reported to cause heart and liver damage, erythrocyte fragility, lethargy, and even death when ingested [5]. Aside from its hazardous effects on health, gossypol is also a known anti-nutritional agent. It can bind proteins, important amino acids, and minerals and interfere with the conversion of feed protein to meat protein in older cattle. These properties of gossypol put restrictions on the use of cottonseed meal as feed, even for ruminants [6].

High-throughput technologies that include solvent extraction or mechanical fractionation, liquid cyclone processing, and adsorption have been used to remove or reduce gossypol and make cottonseed-derived food and feed products safer and more palatable [7]. The application of these technologies, however, is expensive and can significantly increase production costs. The lack of technical advancements in gossypol extraction methods, combined with the high cost of processing, has constrained the use of cottonseed byproducts, particularly in developing countries. In the long term, breeding for tolerable levels of gossypol content in seeds will be the most economical approach to enhance the utilization of cottonseed byproducts as well as to sustain cotton production in the face of worsening agro-environments. To this end, the identification of novel sources of genetic variation for lower gossypol content would be a key step in developing cotton varieties with this important target trait.

Landraces, or traditional varieties of upland cotton (*Gossypium hirsutum*, AADD), are genetically diverse population complexes that remain largely untapped for crop improvement [8]. In the United States, the National Cotton Germplasm Collection (NCGC) maintains about 3420 landraces of cotton [9]. Unlike cultivated crops that have been intensively selected for productivity in highly managed environments, landraces have been selected for their adaptation to seasonal environments within specific localities. This allowed landraces to maintain genetic heterogeneity, making them suitable donors of allelic diversity [8].

The successful use of landraces to add value to a crop in the form of resistance to biotic and abiotic challenges as well as improved agronomic performance has been reported in numerous crops, including rice and wheat [10,11,12]. In cotton, landraces have been shown to be viable genetic resources that can improve tolerance to drought and cold stresses [13,14]. Also, comparisons between cultivated and semi-domesticated accessions of cotton demonstrated that the latter have beneficial alleles, which can enhance fiber improvement and stress tolerance [15,16]. In the past, the incorporation of landraces in cotton breeding programs has been restricted due to their photoperiod sensitivity, negative trait association, and lack of phenotype information [9]. Nevertheless, the development of day-neutral conversion lines of landraces is making it more feasible to incorporate landraces in cotton breeding programs.

In this study, we used genetics and genomics tools to mine quantitative trait loci (QTLs) regulating traits of agronomic interest in Hopi, an allotetraploid (AADD) cotton landrace traditionally cultivated by the Hopi Indians in northeastern Arizona. We identified novel QTLs underlying various agronomic traits including the characteristic naked seed phenotype and reduced glanding in Hopi.

## 2. Results

### 2.1. Agronomic Traits Conferred by the Hopi Allele in the F_2_ Individuals Exhibit Both Mendelian and Non-Mendelian Inheritance

Phenotypic segregation patterns in the F_2_ population show the qualitative nature of stem color and the quantitative inheritance of the remaining traits (Table 1). The segregation ratio for stem color in the F_2_ individuals suggests that the trait was controlled by a single dominant gene with incomplete dominance. Plant height, fiber development, boll glanding, pitting and shape, bract teeth, boll distribution, and growth habit exhibited a continuous distribution in the F_2_ population.

Microscopic examination showed statistically lower gland density in the ovules and the leaves of Hopi compared to TM-1 (Figure 1).

### 2.2. High-Density Linkage Map Displays Co-Linearity with Physical Marker Location

Of the 63,058 SNPs used for genotyping, 7869 (12.48%) lacked allele information, whereas 38,166 (60.53%) were monomorphic. These left 8449 (13.40%) informative markers, of which 8222 mapped to the reference genome (*G. hirsutum* (AD1) ‘TM-*1*’ CRI_v1) (www.cottongen.org (accessed on 7 May 2022)). Specifically, 5143 SNPs mapped to the D sub-genome; 3079 mapped to the A sub-genome; and 848 mapped to both the A and D sub-genomes. Among the SNPs that mapped in the A and D sub-genomes, 28.5% and 32.7%, respectively, showed distorted segregation, i.e., markers deviated from the 1:2:1 expected Mendelian genotypic ratio in an F_2_ population. Interestingly, 18.4% and 25.8% of the markers in the A and D sub-genomes, respectively, segregated in favor of Hopi alleles.

In constructing the linkage map, SNPs with distorted segregation, as well as those occurring in duplicates, were excluded to avoid expanding the length of the linkage groups or increasing the frequency of double crossovers. This resulted in mapping a total of 1836 and 1828 SNPs to the 13 chromosomes of the A and D sub-genomes, respectively. Thirteen linkage groups were assigned for both sub-genomes, with cumulative genetic distances of 2253.11 cM in sub-genome A and 1923.20 cM in sub-genome D. The linkage distance for a single chromosome in sub-genome A ranged from 74.05 cM (A04) to 268.98 cM (A11), with an average marker density of 1.23 cM. In the D sub-genome, the linkage distance ranged from 89.11 cM (D10) to 256.98 cM (D02), with an average marker density of 1.05 cM (Table 2).

The physical location of markers in both sub-genomes showed strong co-linearity with their genetic map positions (Figure 2), although minor inconsistencies in the physical and genetic maps of chromosomes A08 and A12 were observed. The genome coverage provided by the SNPs that mapped in the A and D sub-genomes ranged between 91.41% and 91.16%, respectively (Appendix A).

### 2.3. Hopi Is a Potential Source for the Major QTLs Regulating Traits of Agronomic Interest

A total of 21 QTLs regulating nine traits of agronomic interest were mapped using the segregating population derived from the TM-1 × Hopi cross (Table 3; Figure 3). One QTL each was identified for plant height (*qPH_A11*), bract teeth (*qBRT_A13*) and boll number (*qBN_A01*); two for stem color (*qSC_D3* and *qSC_A13*), boll pitting (*qBPIT_A8* and *qBPIT_A13*) and naked-seededness/fuzz development (*qNS_D12* and *qNS_D13*); three for boll shape (*qBS_A10*, *qBS_A11* and *qBS_D02*); four for boll point (*qBPNT_A13*, *qBPNT_D10*, *qBPNT_D12* and *qBPNT_D13*); and five for gossypol content (*qGC_A11.1*, *qGC11.2*, *qGC_A12*, *qGC_D11* and *qGC_D12*). No QTLs were identified for growth habits. While the PVE of the mapped QTL provides an estimate of the effect of the QTL on the expression of the trait, AE is determined by the marker’s estimated additive effect, with positive AE adding and negative AE subtracting from phenotypic traits. Out of the 21 QTLs identified, five, i.e., *qBPNT_D12*, *qBS_D02*, *qBS_A10*, *qBS_A11*, and *qGL_D12*, have minor effects, while 16 have major effects on the respective phenotypes they are associated with based on recorded PVE values (≥8). A total of 19 out of 21 QTLs were novel, as *qGL_12A* and *qGL_D12* overlapped with two previously reported genes positively regulating gland formation, i.e., *ACGF3* and *DCGF3*, respectively.

The remaining three QTLs identified to regulate boll glanding, *qGC_A11.1*, *qGC_A11.2*, and *qGC_D11*, have not been previously reported and are therefore considered novel (Table 3). *qGC_12A* is located between markers i39206Gh (95,743,855 bp) and i35296Gh (96,426,203 bp) and covers 57 gene annotations. *qGC_D12* is flanked by markers i41961Gh (56,400,783 bp) and i33633Gh (57,932,905 bp) and contains a total of 167 annotated genes.

### 2.4. Transcription Factors Present within the Novel QTLs Regulate Gland Formation

Of the three novel QTLs regulating boll glanding, *qGC_A11.1* was the shortest at 380 kb, followed by *qGC_A11.2* at 570 kb, and *qGC_D11* at 2010 kb. *qGC_A11.1* is flanked by the SNP markers i06723Gh (1,471,933 bp) and i40399Gh (1,849,909 bp) and covers 47 annotated genes, three of which are alternate transcripts. *qGC_A11.2* is located between the SNP markers i26797Gh (11,449,270 bp) and i01021Gh (11,566,012 bp) and contains 36 gene annotations, with one alternate transcript. *qGC_D11* is delineated by markers i19386Gh (68,197,853 bp) and i07687Gh (70,206,382 bp) and consists of 130 gene annotations, 21 of which are alternate transcripts. A total of 213 annotated genes were present within the three QTLs, of which twenty coded for unknown proteins and hence were excluded from downstream in silico analysis. Gene ontology (GO) analysis classified the remaining 193 genes under the aspects of biological processes (BP), cellular component (CC), and molecular function (MF), with considerable overlaps (Figure 4a). The number of genes present within the GO terms under each aspect varied from 21 in the case of catalytic activity (GO: 0003824) to 44 in the case of cellular processes (GO: 0009987) and intracellular anatomical structure (GO: 0005622) (Figure 4a). The thirteen genes that have been previously reported to regulate terpenoid biosynthesis are grouped under the GO terms cellular processes (GO: 0009987), intracellular anatomical structure (GO: 0005622), membrane (GO: 0016020), organelle (GO: 0043226), binding (GO: 0005488), and catalytic activity (GO: 0003824). These genes code for basic helix-loop-helix (*bHLH*; *bHLH1*, *bHLH2*, *bHLH3*, *bLHLH4*), Myb-binding (*MYB*; *MYB1*, *MYB2*, *MYB3*, *MYB4*), zinc finger (*ZF*; *ZF1* and *ZF2*), nicotinamide adenine dinucleotide phosphate (NADP)-binding (*NADP1* and *NAPD2*), and ROSS proteins.

Parallel InterPro analysis grouped the same 193 genes into 15 classes. Three of these classes, namely the NADP-binding domain (IPR 016040), the Myc-type bHLH domain (IPR 011598), and the Myb-domain (IPR 017930), encode proteins related to terpenoid synthesis in cotton (Figure 4b). Eleven genes, namely *bHLH1*, *bHLH2*, *bHLH3*, *bHLH4*, *MYB1*, *MYB2*, *MYB3*, *MYB4*, *NADP1*, *NAPD2*, and *ROSS*, are grouped under these three IPR classes. These genes were included in the thirteen that grouped within the six GO terms related to terpenoid synthesis.

The results of In silico analysis, together with the review of the reported functions of all 193 annotated genes, identified a total of 13 candidate genes regulating boll glanding in cotton.

### 2.5. Comparative Sequence Analysis Identifies Mutations in the Coding Sequences of Candidate Genes Regulating Gland Formation in Hopi

In general, the expression levels of the thirteen candidates identified from the In silico analysis were lower in ovules compared to the leaves (Figure 5a,b). Except for *MYB1*, the relative expression of all candidate genes was uniform (ΔCq = 1) in the ovules and leaves of Hopi but was variable in TM-1 (Figure 5a,b). The expression levels of *bHLH2*, *bHLH4*, *MYB1*, *MYB2*, *MYB3*, and *ZF1* were lower in Hopi ovules compared to TM-1 (Figure 5a), with *MYB2* and *ZF1* showing significant differential expression between the two genotypes. *MYB1* expression was not detected in the Hopi ovules. In the leaves, only *NADP1* and *ZF2* showed higher expression in Hopi (Figure 5b). Interestingly, *MYB2* and *bHLH1* expression in the leaves was 1479 and 80 times lower in Hopi compared to TM-1 (Figure 5b).

Upland cotton is an allotetraploid and is therefore likely to have two or more copies of the same genes within its A and D sub-genomes. In silico analysis identified 11 paralogs of the 13 candidate genes across the A and D sub-genomes. In contrast to the candidate genes, the paralogs had lower expression in the leaves compared to ovules (Appendix A). All the paralogs had reduced expression in the Hopi ovules except the paralogs of *bHLH2* and *MYB4* (Appendix A). The *MYB3* paralog showed the lowest expression in the Hopi ovule. In the leaf tissue, the paralogs had lower expression in Hopi for *bHLH3*, *MYB1*, *MYB3*, *NADP1*, and *ZF1*, while the expression of the remaining paralogs were upregulated (Appendix A).

To confirm the novelty of QTLs identified in this study to regulate boll glanding, comparative expression profiling of three previously reported genes controlling the same trait (*CGF1*, *CGF2*, and *CGF3*) [6] was performed using TM-1 and Hopi ovules and leaves. The reported *CGF* genes mapped to chromosome 12 of both sub-genomes of upland cotton. The expression level of *CGF1* in the ovule was reduced in Hopi, while both *CGF2* and *CGF3* genes were upregulated. This is in contrast to the reported role of these genes in boll glanding (Appendix A). Moreover, *CGF1* and *CGF3* expressions were not detected in TM-1 leaves, while *CGF2* expression was upregulated in Hopi ovules (Appendix A). The expression patterns of the *CGF* genes suggest their inactive role in boll glanding in Hopi.

To identify potential mutations responsible for reduced glanding, comparative analysis of the open reading frames and amino acid sequences was performed for three candidate genes (*MYB2*, *ZF1*, and *bHLH1*), which showed the highest differential expression levels in TM-1 and Hopi. CDS sequencing identified single base mutations and/or insertion/deletion that explains the differential expression levels of the genes (Figure 5; Appendix A). Four SNP mutations were identified in the CDS of the Hopi *MYB2*, with the SNP at the 217th position (G > T) leading to an early stop codon at the 73rd amino acid sequence (Figure 5c). Hopi *bHLH1* had eight SNP mutations, resulting in seven amino acid substitutions. A two base pair (AA) insertion at the 1874–1875th position led to an early stop codon at the 625th amino acid residue. *ZF1* was highly mutated in Hopi, with a total of 15 SNPs. Out of the 15 SNPs, 12 led to amino acid substitutions, with a G > A at the 39th position leading to an early stop codon at the 13th amino acid position.

## 3. Discussion

The narrow genetic diversity in upland cotton cultivars due to generations of industry-scale cultivation of genetically related genotypes, along with the worsening of agro-environments, are continuing threats to cotton production [8,17]. Landraces are reservoirs of novel alleles that can enrich the gene pool of existing cotton cultivars and possibly enhance the agronomic performance of the crop [18]. In this study, we demonstrated the potential of landraces as sources of novel variations by mining Hopi for QTLs that regulate important traits contributing to fiber and seed yield and quality, such as plant height, boll glanding and number, and fiber development.

Hopi, which was previously classified as *G. hopi*, is genetically heterogeneous and therefore amenable to trait improvement through selection [19]. True-breeding lines of Hopi with smooth, pitted, or glandless bolls have been isolated through generations of selection for the traits. In fact, it was the glandless strain of Hopi that has been instrumental in the identification of the loci *gl2* and *gl3* regulating gland formation in cotton [20]. Genetic variations for bract type, stem color, gossypol content, and boll point have also been observed in Hopi populations [19].

Using the 63K SNP set for cotton, we were able to assemble a high-density linkage map with an average marker interval of 1.14 cM. The quality of the linkage map generated in this study was in accordance with other linkage maps constructed using different cotton populations but the same SNP array [21,22]. The high map resolution generated in this study is attributed to higher marker polymorphisms caused by the genetic divergence between TM-1 and Hopi. Chromosomes A12 and D10 had a wider marker interval of 3.64 and 3.07 cM, respectively (Table 2), due to the lower number of polymorphic markers that mapped in these chromosomes. Nevertheless, a strong overall collinearity between the current genetic map and the sequence-based physical map in terms of marker order was established. The minor irregularities in the marker order synteny in chromosomes A05, A08, A12, and D05 are possibly due to the fact that the linkage map was constructed based on recombination events arising from the Hopi × TM-1 genome while the physical location of the markers is based solely on the TM-1 reference genome (Figure 2). The constructed genetic map spans a total of 4185.32 cM, which corresponds to 2033.21 Mb (approximately 91.28%) of the *G. hirsutum* genome sequence (Build CRI_v1) (Appendix A).

Using the generated linkage map, a total of 21 QTLs were identified for nine morpho-agronomic traits. Database searches show that out of these 21 QTLs, 19 have not been previously reported and are therefore novel (https://www.cottongen.org/tripal_megasearch (accessed on 7 May 2022)). These QTLs hold promise for increasing diversity in the cotton gene pool for agronomic and value-added traits. For instance, plant height, which determines plant architecture and yield parameters, also impacts planting density, fruiting branches, and lodging resistance. Traits like glanding, bract type, and stem color serve as indicators of the crop’s stress resilience. In particular, green-stemmed cotton has been reported to be less susceptible to the cotton armyworm, while frego-bract types have been established to provide natural resistance to pink bollworm and boll weevil [23,24]. More economically important traits, such as boll glanding, limit the use of cottonseed as feed, while reduced fuzz fiber can improve the ginning efficiency of cotton fiber [25].

Upland cotton is mainly grown for fiber, but its seed byproducts have found practical applications as alternative sources of edible oil and animal feed. Despite the alternative uses for cotton seeds, they remain underutilized due to the presence of gossypol in the glands. Previous findings have established the positive correlation between glanding and gossypol content in cotton [6,26]. Research into the chemical composition of lysigenous glands identified the predominance of the sesquiterpenoid gossypol along with other aldehyde terpenoids [27]. More recently, the knock-down of *CGF3*-regulating gland formation in upland cotton simultaneously reduced gland formation and gossypol content in the boll surface as well as in the seeds [6]. Associations between gland formation and gossypol content in cotton have been classified as follows: (1) glandless plants—no gossypol; (2) glanded plants with gossypol in seeds, root, stem, and leaves; (3) delayed gland morphogenesis—low or no gossypol; (4) glanded seeds have only low gossypol content; and (5) few glands—low gossypol content [28]. Based on these associations, TM-1 can be classified under category 2, whereas Hopi belongs to category 5.

Of the five QTLs identified to control gland formation in Hopi, QTLs *qGL_12A* and *qGL_D12* overlapped with *ACGF3* and *DCGF3*, respectively, two genes that have been previously validated to control the same trait in near-isogenic lines of Stoneville 7A glandless. *CFG3* genes encode a *bHLH* transcription factor and show differential expression in embryos of glanded vs. glandless cotton 14 days post-anthesis [6]. RNA silencing of *CGF3* has been shown to reduce glanding by 90%, accompanied by a 16-fold decrease in gossypol production, depending on the mutant lines. The *DCGF3* gene (56,642,757 to 56,642,751 bp) maps within the QTL *qGL_D12*, while *ACGF3* (97,044,395 to 97,048,535 bp) lies in close proximity to *qGL_A12*. Our QTL mapping results agree with those obtained from the RNA-seq analysis in the study conducted by Janga et al. in 2019. Expression analysis of the functionally validated CGF genes present in *qGL_A12* and *qGL_D12*, however, did not indicate major roles of these genes in gland formation in Hopi (Appendix A).

Conversely, the three novel QTLs in A11 (two) and D11 (one) showed a strong association with the trait. Gene expression profiling of the 13 candidate genes within these QTLs narrowed down the causal genes into three, namely *bHLH1*, *MYB2*, and *ZF1*.

Gossypol, like most other terpenoids, is a derivative of isopentenyl diphosphate (IPP) arising from the mevalonate (MVA) pathway [29]. IPP yields a large number of terpenoids that can be divided into three major groups, including monoterpenes (ex. essential oils), sesquiterpenes (ex. gossypol, ubiquinone, and prenylated proteins), and diterpenes (gibberellins, chlorophyll, and carotenoids) [29]. The downstream terpenoid products of the MVA-pathway in different crops have been reported to be regulated by transcription factors such as *bHLH*, *MYB*, and *ZF*. *bHLH* has been functionally validated to regulate terpenoid synthesis in several crops, such as *Catharanthus roseus*, *Artemisia argyi*, *Medicago truncatula*, *Salvia miltiorrhiza*, and *Apium graveolens* [30]. *MYB* belongs to a large family of transcription factors known to control terpenoid synthesis in *Vitis viniferea*, *S. miltiorrhiza*, *Dendrobium officinale*, and *Osmanthus fragrans* [31,32]. Together, *bHLH* and *MYB* proteins are known to act as homo- or heterodimers that interact with protein products to regulate several regulatory pathways for the synthesis of terpenoids such as anthocyanin [33,34]. In upland cotton, *CGP1* (*MYB*) and *GoPGF* (*bHLH*) have been suggested to form heterodimers to regulate the synthesis of gossypol and other terpenoids. The decreased *MYB2* and *ZF1* expression in Hopi seeds corresponds to the reported roles of the genes in gland formation in upland cotton (Figure 5a,b). Aside from *bHLH* and *MYB*, zinc finger family proteins have also been reported to regulate terpene biosynthesis in several plant species, such as *Solanum lycopersicum*, *C. roseus*, *Nicotiana attenuates*, and *G. arboreum* as well [35,36].

Surprisingly, the relative expression of the paralogs in the ovules was higher than those of the candidate genes (Appendix A), while expression of the paralogs in the leaves except *bHLH*2 remained basal (less than 2.0) for both genotypes. Moreover, the paralogs had higher expression levels in Hopi leaves (except *MYB3*, *NADP1*, and *ZF1*) that do not match the phenotype, suggesting the lack of involvement of paralogs in leaf glanding. The ovules of upland cotton have gossypol as the major terpenoid, while the leaves are enriched with hemigossypolene [37]. The differences in the expression levels of the candidate genes, as well as the paralog in the leaves and ovules, could be due to the differences in the downstream target of the transcription factors included in the study.

In the present study, expressions of *MYB2*, *bHLH1*, and *ZF1* in both ovules and leaves of Hopi were downregulated (Figure 5a,b). Comparative sequence analysis identified mutations in these genes in Hopi that led to early stop codons (Figure 5c; Appendix A). The candidate genes represent a broad family of transcription factors with gene redundancies across the cotton genome [38,39]. Despite the presence of gene duplicates, the presence of early stop codons in the CDS of the candidate transcription factors would have reduced the overall regulation of downstream gene targets, reducing gland formation in Hopi.

## 4. Materials and Methods

### 4.1. Plant Materials and Development of the Mapping Population

The allotetraploid (AADD) cotton genotypes Texas Marker 1 (TM-1) and Hopi were used as parental lines to generate a segregating population for QTL analysis. TM-1 is an inbred line that was developed in the 1970s as a standard cotton genotype for genetic and cytogenetic experimentation [40]. It has since been used to assemble a reference genome for allotetraploid cotton. Hopi is a cotton landrace traditionally cultivated by the Hopi people of northeast Arizona in the United States [19]. The adaptive responses of this landrace to a range of biotic and abiotic challenges have been associated with its autecology.

Artificial cross-pollination between Hopi as a pollen source and TM-1 as the female parent was carried out following a modified water treatment method [41]. The true hybridity of the resulting F_1_s was confirmed by genotyping using SSR markers that differentiated between the two parental genomes (Appendix A). Briefly, whole genomic DNA was extracted from freshly collected leaves of twelve F_1_ plants following a modified CTAB method [42]. SSR targets from all DNA samples were amplified based on a standard protocol for PCR [43]. All amplicons were resolved in 3% agarose gel in 1× Tris-Borate-EDTA buffer.

The confirmed F_1_ plants were maintained in the greenhouse of the Horticultural Gardens of Texas Tech University (TTU) up to full maturity. Mature bolls were hand-harvested and hand-ginned to obtain the F_2_ seeds.

### 4.2. Morpho-Agronomic Evaluation and Genotyping of the Mapping Population

A mapping population composed of 158 F_2_ individuals, along with the parental lines, was evaluated for various morphological and agronomic traits, including plant height, fuzz development, stem color, growth habit bract-type, and boll characteristics (glanding, pitting, number, and point type) (Figure 6). Phenotypic scores for the traits except fuzz development were assigned based on the standard descriptors for cotton (https://www.cottongen.org/data/trait/NCGC_rating_scale (accessed on 10 October 2021)). The seed nakedness based on the amount of remaining fuzz post-ginning was evaluated on a scale of 1–16, where 1–11 represents fuzzy seeds and 12–16 represent naked seeds [44]. All materials were grown and maintained under standard field management at the Quaker farm of TTU.

At the onset of flowering, young leaves from TM-1, Hopi, and all F_2_ plants were collected for DNA extraction using a modified CTAB method. To ensure the suitability of the samples for SNP genotyping, the quality and quantity of the purified DNAs were assessed using the NanoDrop™ One Microvolume UV-Vis Spectrophotometer (ThermoFisher, Waltham, MA, USA). DNA samples at 1 µg/20 µL were sent to the Texas A&M Institute of Genome Science and Society for genotyping using the Illumina-based 63K Cotton SNP Array [45].

Differences in the extent of gland formation in the ovules and leaves were assessed using an Olympus SZ61 stereomicroscope at 100 µm resolution. Gland density was determined based on three fields of view from three biological samples of ovules and leaves that were randomly collected from Hopi and TM-1 plants post-anthesis. Statistical differences in the number of glands were calculated based on a paired *t*-test at *p* < 0.05.

### 4.3. Linkage Analysis and Genetic Map Construction

The raw SNP genotype data were transformed into an ‘ABH’ mapping format where A, B, and H represent the TM-1, Hopi, and heterozygous alleles, respectively. Only polymorphic markers that behaved co-dominantly and amplified in ≥80% of the segregating population, with a minor allele frequency of ≤0.05, were retained for downstream analysis. The markers were subsequently mapped to the chromosomes of the A and D sub-genomes of upland cotton using TM-1 Build CRI v1 as a reference. The mapped SNPs were then curated manually to identify and eliminate markers that increase the frequency of double crossovers and expand the length of the linkage groups. Distorted markers that deviated from the expected 1:2:1 segregation ratio in the F_2_ population were also excluded from the linkage analysis. The distorted markers were identified manually and tested for goodness-of-fit using the chi-square (χ^2^) test at *p* < 0.05 confidence.

Genetic maps were constructed separately for the A and D sub-genomes of upland cotton using JoinMap version 5.0 [46]. The independence log of odds (LOD) was used as a basis to create different linkage groups. The expected number of linkage groups for the markers in the A and D sub-genomes was obtained at LOD values of 8.0 and 7.0, respectively. Genetic distance within linkage groups was calculated using the Kosambi Mapping Function with a recombination frequency of 0.35 at a goodness of fit of 5.0 [47]. The average marker density for each chromosome was calculated by dividing the total genetic distance for specific chromosomes by the total number of markers in each chromosome. The co-linearity between the genetic and the physical map was visualized using Circos −0.69–9 [48].

### 4.4. QTL Mapping for Target Morpho-Agronomic Traits

The mode of inheritance for each trait was determined based on the segregation ratio of the phenotypes in the F_2_ population. The observed inheritance patterns were then validated using the chi-square (χ^2^) test at *p* ≤ 0.1.

QTLs regulating the target morpho-agronomic traits were analyzed by inclusive composite interval mapping (ICIMapping software version 4.2) [49], with a mapping step of 1 cM. A stringent threshold for each morpho-agronomic trait was set based on 1000 permutation tests. The additive effect (AE) of alleles in a QTL region was based on the estimated additive effect of the marker. Positive and negative AE indicate that the alleles add or subtract the value of a phenotypic trait, respectively. The calculated phenotypic variation explained (PVE) for a given QTL gives an estimate of phenotypic variation due to minor loci with large effects for a given location. A PVE threshold of 8% was chosen as the selection criteria to differentiate minors from the major QTLs.

### 4.5. In silico Identification of Candidate Genes Regulating Gossypol Content in Hopi

In silico analysis of the three novel QTLs identified in this study to regulate boll glanding in Hopi was conducted to identify candidate genes controlling the trait. Details of the annotated genes within each QTL, including gene IDs, map positions and gene descriptions, were retrieved from Cottongen (http://cottongen.org/ (accessed on 11 May 2022)). Genes lacking transcript evidence, as well as those with predicted proteins with unknown functions, were eliminated from downstream analysis. Gene ontology (GO) analysis was carried out to group the remaining genes under GO terms related to gland formation in cotton or terpenoid synthesis. Similarly, InterPro (IPR) analysis was performed to identify protein families that are specific to gland formation and terpenoid synthesis. Both IPR and GO analyses were conducted using the online cotton database CottonFGD (https://cottonfgd.net/analyze/ (accessed on 22 May 2022)). Based on the results of GO and IPR analysis, the available functional annotations for genes within each QTL, as well as the reported functions of each gene across different model plant species, identified candidate genes regulating glanding in Hopi.

### 4.6. Candidate Gene Validation by RT-qPCR and Comparative Sequence Analysis

Comparative expression analysis of the candidate genes and their paralogs in the cotton genome was conducted using leaves and ovules collected at the vegetative stage and 14 days after pollination using gene-specific primers (Appendix A). Genes regulating gland formation in ovules are initiated 14 days post-anthesis (DPA), hence the timing of the collection for this sample [6]. RNA was extracted from three biological replicates of each tissue sample using a modified, hot borate extraction method. Briefly, tissue samples were ground in 3 mL of hot borate extraction buffer (85 °C) supplemented with 60 µL of proteinase K. Finely ground tissues were filtered through a Qiagen spin column at 13,000 rpm for 1 min. The supernatant (1 mL) was transferred to a microcentrifuge tube, supplemented with an equal volume of lithium chloride (LiCl), and kept overnight at 4 °C to maximize RNA yield. The mixture was then centrifuged at 10,000 rpm for 20 min to precipitate the RNA. The extracted RNA was used to synthesize cDNA using the verso cDNA synthesis kit (ThermoFisher Scientific, Waltham, MA, USA). The relative expression levels of the candidate genes in TM-1 and Hopi were determined using the Bio-Rad CFX96 Touch Real-time PCR detection system (Bio-Rad) and analyzed using the Bio-Rad CFX Maestro software version 2.0. Normalization of the relative quantities of candidate genes was carried out using a cotton-specific ubiquitin gene, *UBQ7*.

To confirm the novelty of the identified QTLs in Hopi, the expression of genes that have been previously identified and functionally validated to regulate gland formation, i.e., cotton gland formation (*CGF*) 1, *CGF2*, and *CGF3*, was also analyzed in TM-1 and Hopi following the previously described procedure.

Genes with significant differential expression in TM-1 and Hopi were sequenced and compared to identify genetic variations that are potentially responsible for the observed differences in boll glanding. The entire sequence of candidate genes was amplified following a standard protocol for PCR and using gene-specific primers designed based on available sequence information from Cottongen. Unpurified amplicons were sent to AZENTA Life Sciences for sequencing using the Sanger method. After manually validating the base calls, sequence data for each gene were aligned to that of a reference using CLUSTALW integrated into the BioEdit software version 7.2 [50]. Genes exhibiting sequence variations between TM-1 and Hopi were transcribed and translated In silico using the Nucleic Acid Converter tool (https://skaminsky115.github.io/nac/index.html (accessed on 8 December 2022)). Changes in the amino acid sequence of candidate genes due to nucleotide sequence polymorphisms were identified using CLUSTALW alignment.

## 5. Conclusions

Hopi is a landrace of upland cotton that has gone through partial domestication, thereby retaining higher genotypic and phenotypic variation compared to cultivated varieties. Along with its potential as a rich source of allelic variation, Hopi has characteristic traits such as low levels of glanding and a naked-seed phenotype that can add agronomic value to cultivated cotton. In this study, we highlight the potential of Hopi as a source of genetic variation for several agronomic traits with the identification of 19 novel QTLs for nine different phenotypes. In particular, we identified three new loci that control low glanding density in the bolls of Hopi, which is highly associated with its gossypol content. The results of our study provide a basis for the functional validation of the candidate genes as well as for the analysis of the molecular mechanisms underlying the trait. Advances in our understanding of the genetic and molecular underpinnings of boll glanding have the potential to facilitate genetic and biotechnological manipulation of glanding density in cotton. This will have significant implications in our ability to utilize seed byproducts as a source of edible oil and animal feed without compromising the ability of the cotton plant to respond to biotic stresses.

## Figures and Tables

**Figure 1 ijms-25-07080-f001:**
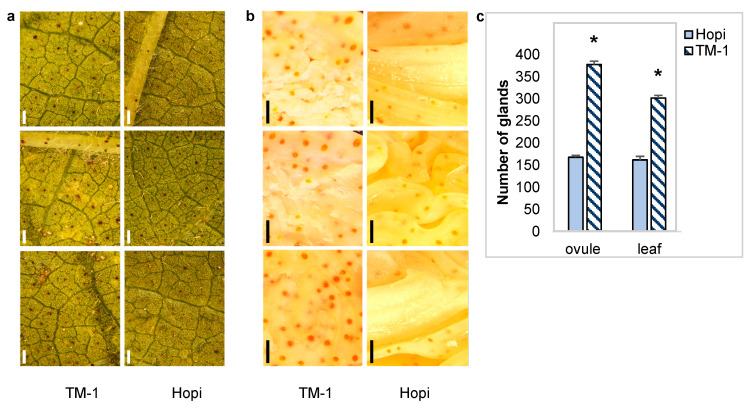
Distribution of glands in the ovules and leaves of Hopi and TM-1. (**a**,**b**) Representative microscopic images at 4.1X magnification showing the distribution of the glands in the ovules and leaves of Hopi and TM-1, respectively. (**c**) Number of glands observed in the ovules and leaves of Hopi and TM-1; asterisks (*) indicate significant differences in glanding density at *p* < 0.05 confidence. bar = 100 µm in (**a**,**b**).

**Figure 2 ijms-25-07080-f002:**
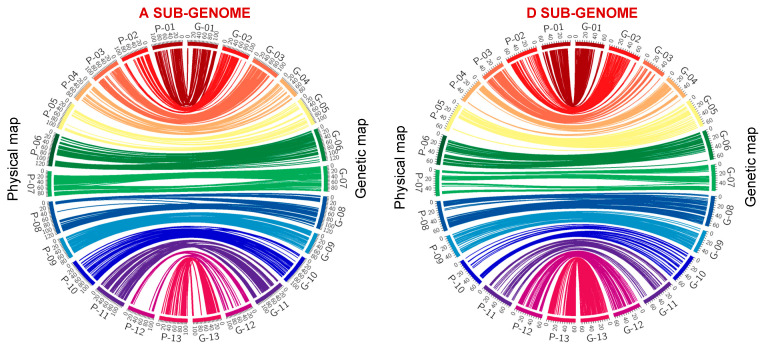
Circos map showing co-linearity between the genetic map and physical map. P01–P13 and G01–G13 represent marker locations in the physical and genetic map, respectively.

**Figure 3 ijms-25-07080-f003:**
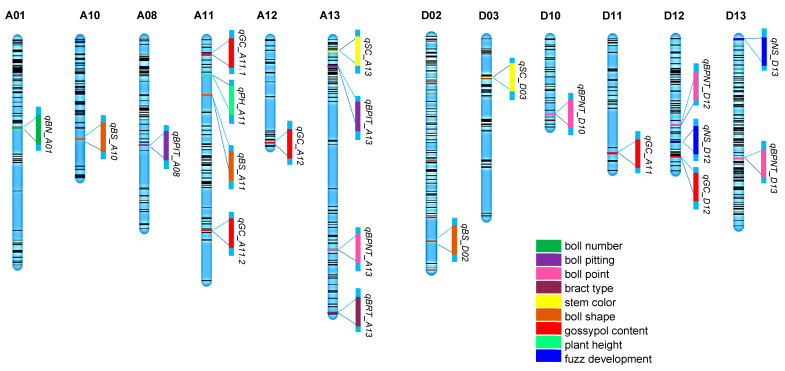
Map position of the 21 QTLs regulating various traits in Hopi. Chromosomes are represented by blue bars, and black lines are markers based on the genetic map.

**Figure 4 ijms-25-07080-f004:**
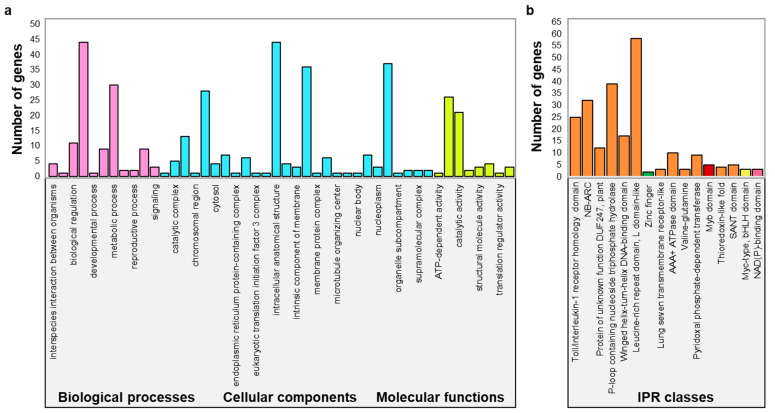
In silico analysis of the transcripts present within the novel QTLs. (**a**) Gene ontology classification of genes present within the novel QTLs regulating gland formation. (**b**) InterPro classification of genes within the novel QTLs regulating glanding; the pink, yellow, red, and green bars represent IPR groups containing genes involved in gland formation or terpenoid synthesis in cotton and other plant species.

**Figure 5 ijms-25-07080-f005:**
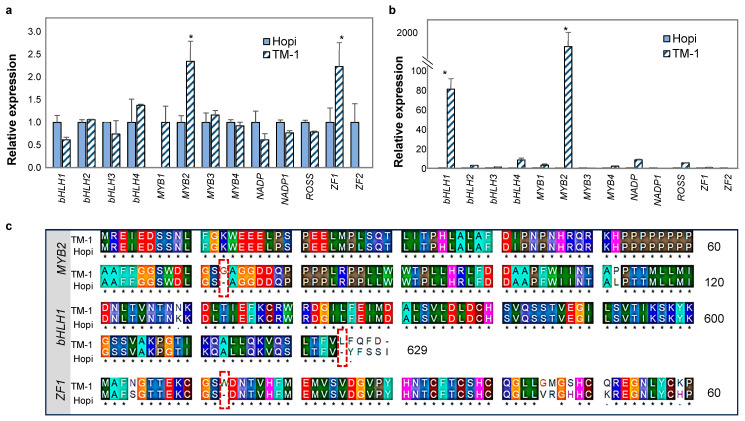
Comparative expression and sequence analysis for the candidate genes. (**a**,**b**) Relative expression profiling of candidate genes in tissue samples from ovules and leaves of TM-1 and Hopi, respectively. Asterisks (*) indicate statistically significant differential expression of the candidate genes between Hopi and TM-1. (**c**) Representative comparative amino acid sequence analysis of the candidates *MYB2*, *bHLH1*, and *ZF1*. Numbers on the **right** represent the amino acid residues. Red boxes highlight the mutation events that lead to early stop codons in the Hopi.

**Figure 6 ijms-25-07080-f006:**
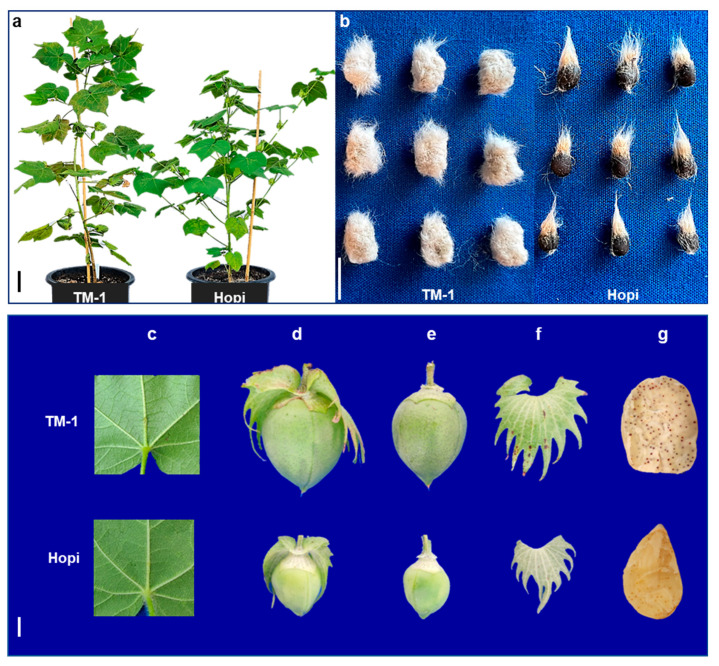
Variations in morphology of the parental genotypes. (**a**) Plant height, stem color, and growth habit; (**b**) Fuzz characteristics; (**c**) Leaf glanding; (**d**–**f**) Boll and bract characteristics; and (**g**) Boll glanding of TM-1 and Hopi. bar = 10 cm in (**a**) and 1 cm in (**b**–**e**).

**Table 1 ijms-25-07080-t001:** Inheritance of morpho-agronomic traits based on phenotypic distribution in F_2_ individuals.

Trait	Parental Phenotype		Segregation Ratio in F_2_	χ^2^ Test	Inheritance
	TM-1	Hopi			
Plant height	35.5 cm	26.9 cm	continuous	-	non-Mendelian
Stem color	red	green	1.2:2.0:1.0	1.06 ^a^	Mendelain
Gossypol formation	heavy	light	continuous	52.73	non-Mendelian
Boll pitting	intermediate	smooth to light	continuous	28.88	non-Mendelian
Boll shape	moderate	pointed	continuous	106.7	non-Mendelian
Bract teeth	medium	small	continuous	3.97 ^a^	non-Mendelian
Boll Distribution per node	1	1–4	continuous (1–4)	1.68 ^a^	non-Mendelian
Growth habit	compact	prostate	continuous	51.33	non-Mendelian
Fuzz development	fuzzy	naked	continuous	-	non-Mendelian

^a^ indicates that the observed phenotypic distribution was tested against the Mendelian ratio at a 0.05 level of confidence.

**Table 2 ijms-25-07080-t002:** Distribution of the polymorphic SNP markers among the 13 linkage groups for each sub-genome.

Chr ^a^	A Sub-Genome			D Sub-Genome		
	No. of Markers	Chr ^a^ Length (cM)	Marker Density	No. of Markers	Chr ^a^ Length (cM)	Marker Density
1	162	176.2	1.09	284	122.01	0.43
2	148	255.54	1.73	136	256.98	1.89
3	150	160.23	1.07	158	176.97	1.12
4	49	74.05	1.51	84	181.94	2.17
5	82	143.61	1.75	84	105.27	1.25
6	97	102.1	1.05	197	152.97	0.78
7	184	159.9	0.87	109	164.39	1.51
8	231	227.54	0.99	147	92.52	0.63
9	152	188.04	1.24	155	130.6	0.84
10	165	207.56	1.26	39	89.11	2.28
11	213	268.98	1.26	44	134.94	3.07
12	39	141.96	3.64	137	132.14	0.96
13	164	147.4	0.9	254	183.36	0.72
	1836 ^b^	2253.11 ^b^	1.23 ^c^	1828 ^b^	1923.20 ^b^	1.05 ^c^

^a^ chromosome; ^b^ indicates the sum of values for the number of markers and for both sub-genomes; ^c^ indicates the average marker density.

**Table 3 ijms-25-07080-t003:** Twenty-one QTLs regulating morpho-agronomic traits identified using composite interval mapping and interval mapping.

QTL Name	Chr ^a^	Left Marker	Right Marker	Marker Range (Mb)	PVE ^b^	AE ^c^
*qPH_A11*	A11	i45381Gh	i45705Gh	5.17–5.21	15.01	−0.74
*qSC_D03*	D3	i23719Gh	i43062Gh	22.35–26.43	16.05	0.45
*qSC_A13*	A13	i49771Gh	i31374Gh	6.13–13.21	13.94	0.3
*qGC_A11.1*	A11	i06723Gh	i40399Gh	1.47–1.85	9.61	−0.15
*qGC_A11.2*	A11	i26797Gh	i01021Gh	114.49–115.06	10.34	−0.1
*qGC_D12*	D12	i08451Gh	i21972Gh	56.40–57.93	2.28	−0.16
*qGC _A12*	A12	i39206Gh	i35296Gh	95.74–96.43	8.88	−0.37
*qGC _ D11*	D11	i19386Gh	i07687Gh	68.2–70.21	12.97	−0.13
*qBPIT_A8*	A8	i52750Gb	i50859Gb	120.85–122.26	9.69	0.26
*qBPIT_A13*	A13	i28698Gh	i35418Gh	27.28–29.74	9.02	0.23
*qBPNT_A13*	A13	i13421Gh	i13455Gh	91.75–94.84	14.46	0.37
*qBPNT_D12*	D12	i32198Gh	i08128Gh	44.79–45.17	1.27	−0.27
*qBPNT_D10*	D10	i12182Gh	i17593Gh	59.95–60.12	10.44	0.36
*qBPNT_D13*	D13	i36840Gh	i29868Gh	49.59–51.22	10.58	0.37
*qBRT_A13*	A13	i63505Gm	i13876Gh	107.32–108.52	12.22	0.37
*qBS_A10*	A10	i407935Gh	i43538Gh	105.90–113.48	5.43	0.37
*qBS_A11*	A11	i45381Gh	i45705Gh	5.08–17.39	6.45	−0.47
*qBS_D02*	D02	i50830Gb	i05555Gh	65.40–65.70	2.09	0.37
*qBN_A01*	A01	i29428Gh	i34358Gh	51.66–52.10	10.11	−0.16
*qNS_D12*	D12	i08313Gh	i51812Gb	52.66–53.06	14.65	−1.26
*qNS_D13*	D13	i41961Gh	i33633Gh	0.18–0.69	38.19	−1.95

^a^ chromosome; ^b^ phenotypic variance explained; ^c^ additive effect.

## Data Availability

All sequence data are deposited in NCBI under the GenBank accessions OQ417761, OQ417762, OQ417763, OQ417764, OQ417765, and OQ417766.

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
