# Peer review of "Genetic Analysis of an F2 Population Derived from the Cotton Landrace Hopi Identified Novel Loci for Boll Glanding"

_ijms, 2024, doi:10.3390/ijms25137080_

Round 1
Reviewer 1 Report
Comments and Suggestions for Authors
The manuscript is prepared on a current and interesting topic. However, it contains deficiencies that must be corrected or supplemented before its acceptance.
Introduction – paragraph (72-80) should be added about the importance of ploidy in cotton, which plays an important role in the genetics and breeding of cotton, as an example I would cite the study by Revanasiddayya et al. (2024, DOI: 10.17221/12/2023-CJGPB). Especially if the authors work within the results with sub-genomes A and D (this issue is also discussed). Similarly, lines 72-74 show the importance of landraces for resistance breeding, but the example is given for rice and wheat. There is also a study by Gu et al. (2023, DOI: 10.17221/08/2022-CJGPB) dedicated to the study of the cytochrome P450 gene and drought, so it would be appropriate to add a reference to cotton and not just use wheat and rice.
Results - at first glance they are adequately described, but their evaluation is only possible after the addition of essential methodological information. In its current form, I would add explanations of the used abbreviations to the legend of Table 3. As part of sequence analyses, the results must be supplemented with sequence codes that the authors obtained and entered into the relevant database (e.g. NCBI is the standard today). This insertion can be implemented as a reference to TableS. Without this, they lose their meaning and their discussion is also problematic.
Discussion - the same statement applies as for the results.
Materials and Methods – for the genotypes used, it is appropriate to add the ploidy level and the genome. In the case of SSR analysis (line 499), it is stated that some protocol was used, etc. However, I consider it essential in view of the association study to state the number of SSR markers and their location in the genomes (again, possibly as a Supplementary file), because otherwise it is not possible to adequately evaluate results and their discussion. Similarly, subsection 4.6 dedicated to RT-qPCR lacks information on the primer combinations used in the study. Again, essential information is missing, which must be supplemented (e.g. in the form of TableS).
References – there are different formats for citing journals, i.e. abbreviated and full (sometimes with misspelled upper and lower case letters) - must be unified according to the requirements for authors.
Supplementary files – Fig. S1 – the legend lacks a description of the line segments indicating the variability and possibly a statistical analysis, because there are possible statistical differences. Table S1.1 and S1.2 - wrong number format (does not correspond to English standards).
Based on the above, I recommend the manuscript for publication after major revision and second review.
Author Response
Reviewer #1:
The manuscript is prepared on a current and interesting topic. However, it contains deficiencies that must be corrected or supplemented before its acceptance.
Comment 1: Introduction – paragraph (72-80) should be added about the importance of ploidy in cotton, which plays an important role in the genetics and breeding of cotton, as an example I would cite the study by Revanasiddayya et al. (2024, DOI: 10.17221/12/2023-CJGPB). Especially if the authors work within the results with sub-genomes A and D (this issue is also discussed). Similarly, lines 72-74 show the importance of landraces for resistance breeding, but the example is given for rice and wheat. There is also a study by Gu et al. (2023, DOI: 10.17221/08/2022-CJGPB) dedicated to the study of the cytochrome P450 gene and drought, so it would be appropriate to add a reference to cotton and not just use wheat and rice.
Answer 1: The importance of ploidy was excluded from the introduction section due to lack of coherence in the focus of the section. The concept of ploidy level for the target trait is described in terms of comparison between the paralog genes the candidates in lines 439-442 of the original manuscript.
For lines 72-74 the focus is to put forth a general emphasis on landraces as important resources that can be used for crop improvement and hence rice and wheat as examples. Wheat and rice are two of the big three in cereals, that’s why those papers were cited. In the study by Gu et al. (2023, DOI: 10.17221/08/2022-CJGPB) dedicated to the study of the cytochrome P450 gene and drought, the germplasm used is CCRI74. However, the article does not explicitly classify CCRI74 as a landrace. As suggested, lines 76-77 were added to the updated manuscript to add more references involving cotton landraces.
Comment 2: Results - at first glance they are adequately described, but their evaluation is only possible after the addition of essential methodological information. In its current form, I would add explanations of the used abbreviations to the legend of Table 3. As part of sequence analyses, the results must be supplemented with sequence codes that the authors obtained and entered into the relevant database (e.g. NCBI is the standard today). This insertion can be implemented as a reference to TableS. Without this, they lose their meaning, and their discussion is also problematic.
Answer 2: The lines 165-167 were added to the updated manuscript to add explanations of the used abbreviations to the legend of Table 3.
The results for the sequence analysis have been supplemented with the sequence codes from NCBI as a supplementary table (Table S2) in line 326 of revised manuscript.
Comment 3: Discussion - the same statement applies as for the results.
Answer 3:
Comment 4: Materials and Methods – for the genotypes used, it is appropriate to add the ploidy level and the genome. In the case of SSR analysis (line 499), it is stated that some protocol was used, etc. However, I consider it essential in view of the association study to state the number of SSR markers and their location in the genomes (again, possibly as a Supplementary file), because otherwise it is not possible to adequately evaluate results and their discussion. Similarly, subsection 4.6 dedicated to RT-qPCR lacks information on the primer combinations used in the study. Again, essential information is missing, which must be supplemented (e.g. in the form of TableS).
Answer 4: The ploidy level and the genome level for the genotypes used has been added on line 485 of the updated manuscript.
The supplementary tables, Table S3 (line 504) and Table S4 (line 599), which contain details on SSR markers used in the association study and primer combinations for RT-qPCR, respectively had been included in the revised manuscript. The tables include the marker names, sequence for forward and reverse primes, genomic location and expected amplicon size of the PCR product.
Comment 5: References – there are different formats for citing journals, i.e. abbreviated and full (sometimes with misspelled upper and lower case letters) - must be unified according to the requirements for authors.
Answer 5: The references have been reformatted and unified according to the requirements in the revised manuscript.
Comment 6: Supplementary files – Fig. S1 – the legend lacks a description of the line segments indicating the variability and possibly a statistical analysis, because there are possible statistical differences. Table S1.1 and S1.2 - wrong number format (does not correspond to English standards).
Answer 6: Fig. S1 has been updated. A statistical t-test was performed for the paralog genes. The statistically significant differences (p < 0.05) in expression level of each gene between two genotypes is indicated by asterisks.
The supplementary Tables have been updated and formatted based on the recent publication by Huang et al. 2024 (https://www.mdpi.com/1422-0067/25/12/6363).
Reviewer 2 Report
Comments and Suggestions for Authors
The authors investigated used genetics and genomics tools to mine quantitative trait loci of agronomic interest in Hopi, a cotton landrace traditionally cultivated by the Hopi Indians in northeastern Arizona.
General concept comments
The manuscript is well written and well structured. It is relevant for the field and clear. The experimental design is appropriate, and the results are reproducible. The manuscript is accompanied by high quality figures. Data Availability Statement is correctly reported. References are not correctly reported, they must be changed. They are not recent, and some self-references have been detected. Conclusions section is consistent with results. Minor corrections are needed.
Specific comments
The title is too long.
The introduction must not contain results. I suggest removing or moving lines 83-91.
The introduction can be improved, adding more literature background.
The discussion should be improved. Lines 341-355 report only self-reference 8 and reference 16 (1938).
Some key words can be added.
Comments on the Quality of English LanguageMinor editing of English language required
Author Response
Reviewer #2:
The manuscript is well written and well structured. It is relevant for the field and clear. The experimental design is appropriate, and the results are reproducible. The manuscript is accompanied by high quality figures. Data Availability Statement is correctly reported. References are not correctly reported, they must be changed. They are not recent, and some self-references have been detected. The Conclusions section is consistent with results. Minor corrections are needed.
Comment 1: The title is too long.
Answer 1: We agree with the title being long, however, it was done to encapsulate the key information in the research. The title has been changed to “Genetic analysis of an F2 population derived from the cotton landrace Hopi identified novel loci for boll glanding” to reduce the length of without compromising the essence.
Comment 2: The introduction must not contain results. I suggest removing or moving lines 83-91.
Answer 2: The lines 86-91 on page 2 of the original manuscript have been removed.
Comment 3: The introduction can be improved, adding more literature background.
Answer 3: The introduction was designed to highlight two major aspects of cotton, the economic importance of gossypol in cotton and value of landraces in cotton breeding. The first four paragraphs of the original manuscript, lines 30 to 64, focus on the impact of gossypol in cottonseeds for sustainable cotton cultivation. The fifth and sixth paragraph, lines 64 – 80, in the original manuscript highlight landraces are readily available source for enhancing agroeconomic as well as resilience in the cotton gene pool. The lines 79-77 in the revised manuscript provide further literature background on the use of cotton landraces.
Comment 4: The discussion should be improved. Lines 341-355 report only self-reference 8 and reference 16 (1938).
Answer 4: The focus of our lab is concentrated towards utilization of naturally available genetic resources, crop distant relatives, enhancing the performance of commercially available germplasm. Self-reference 16, is one of the previous publications which sets foundations for the present-day research in the lab. Despite the potential of landrace Hopi as a donor of novel alleles for several traits, intensive study on the genotype has not been performed. The references [17 and 18] have been added to the updated manuscript to make discussion more coherent.
Comment 5: Some key words can be added.
Answer 5: Two key words transcription factors and QTL mapping has been added in the updated manuscript.
Round 2
Reviewer 1 Report
Comments and Suggestions for Authors
The authors have addressed most of my comments. I consider the information that ploidy in cotton was dealt with as part of the discussion to be insufficient settlement. In particular, if we are talking about the use of "landrace" materials, which is the basis for the study, then its ploidy plays a role in basic as well as applied research in the given area. Therefore, I consider the inclusion of a paragraph in the Introduction section to be essential for a better understanding of the entire study. In my opinion, information about ploidy in the range of about 10 lines should be incorporated into the Introduction section (including the suggested reference Revanasiddayya et al. (2024, DOI: 10.17221/12/2023-CJGPB), but there are also many other studies on this issue), e.g. as part of a paragraph (line 72-82).
The other comments are dealt with sufficiently.
Due to the absence of part of the problems in the Introduction section, I propose to accept the manuscript for publication after major revision and second review, because cotton ploidy plays a crucial role in the application of the results described by the authors.